# Metal–Organic Framework-Assisted Atmospheric Water Harvesting Enables Cheap Clean Water Available in an Arid Climate: A Perspective

**DOI:** 10.3390/ma18020379

**Published:** 2025-01-15

**Authors:** Yang Hu, Linhui Jia, Hong Xu, Xiangming He

**Affiliations:** 1Institute of Nuclear and New Energy Technology, Tsinghua University, Beijing 100084, China; byronace@163.com (Y.H.); jlh@hainanu.edu.cn (L.J.); 2School of Chemistry and Chemical Engineering, Hainan University, Haikou 570228, China

**Keywords:** Metal–Organic Frameworks (MOFs), Atmospheric Water Harvesting (AWH), porous materials, sorbent materials

## Abstract

Extracting water directly from the atmosphere seems to be a perfect way to solve the water scarcity facing 2 billion people; however, traditional Atmospheric Water Harvesting (AWH) lacks the ability to adsorb water molecules in an arid climate. Porous materials are capable of assisting water adsorption; however, currently, only certain customizable Metal–Organic Frameworks (MOFs) are able to meet the standard of adsorbing water molecules at low humidity and releasing water at low temperatures at certain times that can realize assisted AWH’s practical and energy-efficient use (Energy consumption < 5kWh/L-water). From this perspective, we offer a concise review of the advancements in enhanced AWH technologies, delve into the attributes of appropriate MOFs, and offer insights into the potential and future directions of MOFs–AWH. In conclusion, we underscore that that the development of designable MOFs holds the key to the widespread practical implementation of AWH, promising the availability of affordable clean water anywhere in the world.

## 1. Introduction

Roughly two-thirds of the globe’s inhabitants are contending with water shortages [1], while an astonishing 43% of the earth’s surface is vulnerable to desertification [2,3]. Moreover, persistent and expansive droughts, exacerbated by the capricious nature of climate change, are restricting access to potable water and increasingly weighing down on human society. Addressing the issue of water scarcity, especially in arid zones, is critically urgent. Water is omnipresent, enveloping us in the form of vapor and droplets suspended in the atmosphere. Studies have suggested that the volume of water vapor in the air amounts to a staggering 12,900 cubic kilometers—nearly one-seventh of the combined volume of the world’s lakes and rivers [4]. If we could harness water directly from the atmosphere [5], the challenges posed by the uneven distribution of freshwater and the dearth of water in arid regions might be substantially mitigated.

In scholarly discourse, it is termed Atmospheric Water Harvesting (AWH), whereas the industry knows it as an Atmospheric Water Generator (AWG). This innovative technique for procuring water entails chilling the air to a temperature significantly beneath its dew point to induce condensation [6,7]. Despite its potential, AWH is seldom utilized in arid regions where humidity is notably low. This reluctance stems not only from the substantial energy demands of the water extraction process, which can vary from around 2.78 to a steep 27.8 kilowatts per kilogram of water [8], but also from the formidable challenge of capturing water molecules in such arid conditions.

Enhanced Atmospheric Water Harvesting, or assisted AWH, represents a technological advancement over the conventional AWH process. It harnesses the capabilities of porous materials that can extract water molecules from the air even in low-humidity environments, thus making it feasible to operate in arid zones [9]. To date, only a handful of designable Metal–Organic Frameworks (MOFs) have proven effective in augmenting the AWH technique [10,11,12,13,14,15,16,17,18,19,20]. In contrast to other reviews or perspectives that focus on enumerating various adsorbent materials or delving into the thermal engineering aspects of assisted AWH, this perspective focuses on the progress of MOFs within assisted AWH and suggests that MOFs have the potential to emerge as the most cost-effective and efficient method for providing fresh water. The perspective commences by charting the development of assisted AWH, continues with an analysis of the properties of promising MOFs, and culminates with the authors’ insights and projections for the future role of MOFs in assisted AWH.

## 2. The Developing Assisted AWH

### 2.1. Principle

Traditional Atmospheric Water Harvesting (AWH) technology relies on the condensation of humid air to separate water from dry air, as depicted in Figure 1A. However, this process becomes notably difficult when the surrounding air is low in humidity. Enhancing the AWH system by integrating water-adsorbent materials upstream to elevate humidity levels can streamline the process [21,22]. The operation of the enhanced AWH process is as follows: (1) porous materials spontaneously adsorb water vapor, locally transitioning humidity from low to high; (2) these water-saturated porous materials are then heated to liberate water vapor; (3) subsequently, the vapor is condensed to gather water, as shown in Figure 1B. To maintain an adequate water supply, the enhanced AWH system must be cycled multiple times daily, which requires the adsorbent porous materials to exhibit four critical attributes: (1) the capacity to adsorb water at low humidity; (2) a substantial water uptake capability; (3) the ability to release water at low temperatures to ensure energy efficiency; and (4) robust cycling stability along with effective water retention properties.

### 2.2. Searching the Suitable MOFs

A variety of water-adsorption porous materials have been tested. Conventional options include mesoporous carbon [41,42], mesoporous silicon [43], zeolites [44], and desiccant silica gel [45], yet experimental results indicate that these materials either fail to adsorb water at low humidity or struggle to desorb it in an energy-efficient manner.

Metal–Organic Frameworks (MOFs) constitute a class of crystalline porous materials, characterized by metal ions or clusters as nodes and organic ligands as linkers [46,47,48,49,50,51,52]. The design flexibility of MOFs holds promise for their application in assisted AWH [26,28,53]. Early-generation MOFs, such as HKUST-1 [54], MOF-5 [55], and MOF-74 [56], either decomposed upon contact with water or adsorbed it only at high humidity (>60%RH), like MCM-41 [57]. Subsequent MOFs, including MIL-101 [58], UiO-66 [59], Co_2_Cl_2_(BTDD) [12], and Ni_2_Br_2_BTDD [17], managed to adsorb water at 20~60%RH, but this still falls short of the requirements for arid climate conditions (15~40%RH). It was not until the development of CAU-10 [60], DUT-67 [61], MOF-801 [62], MOF-303 [37], and MOF-LA2-1 [20] that water adsorption at 20%RH became feasible. For practical applications, factors such as the water adsorption rate, capacity, and cycling stability must be taken into account. To date, MOF-801 and MOF-303 have been the subjects of extensive research as they have excellent water adsorption characteristics at lower humidity and have large water adsorption capacity (Figure 1C,D) [29,31,37,38,40,63,64,65,66]. Recently, novel MOFs like MOF-333 [29], CAU-23 [67], and MOF-LA2-1 [68] have also been explored for their potential in assisted AWH.

### 2.3. Engineering Progress

Over the past seven years, the synergy between material science and engineering has propelled MOF-assisted Atmospheric Water Harvesting (AWH) closer to practical implementation (Figure 1E). In 2017, Omar M. Yaghi’s group pioneeringly demonstrated the collection of water from the air using MOFs, powered by natural sunlight, at a remarkable humidity level as low as 20% [31]. The following year, with the scaling up of the water-harvesting apparatus, a yield of 100 g of water per kilogram of MOF-801 was achieved [37]. Notably, this process leveraged the temperature differential between day and night for a single cycle of water collection. To increase the yield, it is essential to enhance the frequency of water intake cycles. Consequently, active cycling of heating and condensation has been broadly adopted, resulting in the production of over 1 L of water per kilogram of adsorbent per day [38].

## 3. Characteristics of Suitable MOFs

### 3.1. MOF Properties

MOFs constitute a class of porous materials composed of metal nodes and organic linkers. Their modular nature allows for great synthetic tunability, affording both fine chemical and structural control. With creative synthetic design, properties such as porosity, stability, porous size, and porous environment, especially the chemical and physical bond force, can be tailored to specific applications. The pore size and pore environment in MOFs play an important role in the adsorption and desorption process of water. For low-humidity conditions, capturing sparse water molecules requires pores that are close in size to water molecules and have hydrogen-bonding adsorption capabilities. When water molecules aggregate and their volume increases, large pores are needed to accommodate them.

### 3.2. Facing the Challenges

In the water-harvesting process, the capacity to “strongly adsorb water molecules at low humidity” followed by “releasing water at a lower temperature” presents a significant trade-off, constituting a major scientific challenge. Adsorbing water molecules at low humidity suggests that the material has a high affinity for water. When a material forms a strong bond with water molecules, it is challenging to separate them, making it seemingly impractical to release the tightly bound water at lower temperatures.

In principle, the optimal strategy for resolving this trade-off involves decoupling the problem through the segregation of space and time. This entails addressing one issue at a specific moment and location, and then, as conditions change, employing alternative tactics to confront the subsequent challenge. Various issues can be resolved by different problem-solving entities or by distinct components within a single entity. Designable multi-porous Metal–Organic Frameworks (MOFs) act as a single entity with varied components that can effectively navigate the trade-off inherent in enhanced AWH, ultimately facilitating efficient water extraction.

### 3.3. MOFs with More Pores Bring More Hope

Certain designed MOFs feature multiple pores with various functionalities. The small pores (with diameters of nearly 3.4 Å) exhibit strong interactions with individual water molecules, enabling the capture of water at a low relative humidity [16,19,51,69]. As water molecules aggregate into clusters, the larger pores (with diameters of nearly 6.0 Å) provide storage space. At this stage, the interaction between the large pores and the water clusters weakens, allowing water to be desorbed at lower temperatures [29]. Notably, within a single MOF, the transition of water molecules from small to large pores occurs naturally and seamlessly. The multifunctional pores of MOFs, which can alter their properties over time and space, are a feature absent in traditional single-pore materials, satisfying the criteria essential for assisted AWH.

### 3.4. Functionalize Magic Pores

Water contains a large number of hydrogen bonds [70,71]. When MOF’s pores have hydrogen bonds and their diameter is similar to the water molecules (3.4 Å), an individual water molecule can be captured easily. The volume density of hydrogen bonds—the number and proximity of hydrogen bonds within the pores—significantly influences the adsorption and storage of water. The count of hydrogen bonds can be manipulated by altering the prevalence of nitrogen, oxygen, and fluorine elements within the organic ligands that serve as the connectors in MOFs. Additionally, the distance between hydrogen bonds can be adjusted by modifying the length of the organic ligands.

To exemplify how functional pore design addresses the trade-off between “adsorbing water molecules specifically at low humidity” and “releasing them at low temperature”, we can look at the multiple-pore MOFs, such as MOF-303 and MOF-801. These materials illustrate how the manipulation of pore size and environment effectively resolves this challenge.

Figure 2A illustrates MOF-303, constructed with Al^3+^ as the central metal ion and 2,3-pyrazinedicarboxylic acid (H2PZDC) as the ligand, featuring pores with diameters of 3.4 Å and 6.0 Å. Both sizes exceed the diameter of a single water molecule, which is 2.8 Å. The H2PZDC ligand possesses two carboxylic acids (O-H) and a pyridine ring (N-H), leading to two types of hydrogen bonds: O···H and N···H. The 3.4 Å pore wall is adorned with four N-H hydrogen bonds, while the 6.0 Å pore, oriented perpendicularly, contains three O-H bonds and four N-H bonds, totaling seven hydrogen bonds [38]. At low humidity, a single water molecule is readily captured by the high-density hydrogen bonds within the 3.4 Å pore. As depicted in Figure 2(Ba), the first water molecule forms three hydrogen bonds with two pyrazole groups and one μ2-OH group. Subsequent water molecules fill the 3.4 Å pore, as shown in Figure 2(Bb–Bd). The fifth and subsequent water molecules overflow into the adjacent 6.0 Å pore [29], as seen in Figure 2(Be). Consequently, this material boasts a high density of hydrogen bonds in the smaller pores and a moderate density in the larger pores, enabling it to adsorb water at extremely low humidity (RH = 20%). The red S mark in Figure 2A denotes the initial stage of the cycling process. Simultaneously, water desorption occurs at a mere 65 °C, with the adsorption and desorption cycle being completed in less than 25 min [38].

In addition to MOF-303, MOF-801 also stands out as a multifunctional water-adsorbent material. As illustrated in Figure 2C, MOF-801 features pore sizes of 4.8 Å, 5.6 Å, and 7.4 Å and is synthesized using the weakly polar Zr^4+^ as the central metal ion in conjunction with fumaric acid as the ligand. The 4.8 Å pores, equipped with two O-H bonds, are capable of capturing water molecules at low humidity as well. With a pore volume of 0.45 cm^3^ g^−1^ for MOF-801, which is less than that of MOF-303 (0.54 cm^3^ g^−1^), the material’s final water uptake weight percentage reaches 32%, marginally lower than the 42% achieved by MOF-303 [28].

We have compiled the water adsorption capabilities of various materials, as presented in Figure 1D and Table 1. From a physical and chemical standpoint, the efficiency of water adsorption and desorption is influenced by the isosteric heat of adsorption (Q_st_) and the latent heat of water vaporization (40.7 kJ mol^−1^) [28,72]. Adsorption and desorption can only be rapidly interconverted when these two values are closely aligned. The Q_st_ values for UiO-66, MOF-841, and MOF-802 are all around 50 kJ mol^−1^, nearly matching the latent energy of water (40.7 kJ mol^−1^), which is why their adsorption humidity range falls between 20% and 50%, as shown in Figure 1D. Materials with a higher Q_st_ (70 kJ mol^−1^), such as Zeolite-13 and MOF-74-Co, can adsorb water at very low humidity (<10%) and reach saturation swiftly, as depicted in Figure 1D. However, the desorption temperature required for these materials can reach up to 200 °C, which does not align with the standards set for assisted AWH [44].

Upon examining MOF-303 and MOF-801, which have Q_st_ values of up to 60 kJ mol^−1^, slightly lower than 70 kJ mol^−1^, we find that they can adsorb water at low humidity (~10%), suitable for arid climates. Interestingly, their adsorption cutoff is at 35% humidity, which, coincidentally, is the point of steepest water adsorption (i.e., the adsorption/desorption equilibrium point) for porous materials with a Q_st_ of 40 kJ mol^−1^. This suggests that the saturated water adsorption isosteric heat of MOF-303 or MOF-801 has transitioned to 40 kJ mol^−1^. Consequently, the saturated MOF-303 or MOF-801 can release water rapidly with minimal heating.

From this, we can ascertain that multiple pores can address the primary contradictions that arise at different times and in various spaces. Firstly, the high-volume density of hydrogen bonds raises the isosteric heat of adsorption to up to 60 kJ mol^−1^, facilitating the trapping of water molecules at relatively low humidities (~10%). As water accumulates, the water molecules cluster together, weakening the binding force between the water and the pore. This reduction in the isosteric heat of adsorption to the latent energy of water (40.7 kJ mol^−1^) makes it easier to desorb water with minimal energy input. Notably, resolving these issues in different times and spaces does not solely rely on the multi-pore approach; any method that can generate distinct pore environments varying over time and space would be effective.

## 4. Outlook and Perspectives

We will look forward to MOF-assisted AWH from three aspects: water intake cost, usage scenarios, and areas for improvement in the future.

### 4.1. Be the Cheapest Clean Water Provider

Clean water comes at a high cost, with the global average price of bottled water reaching $0.7 per kilogram of H_2_O [74], not to mention the expenses in arid regions. MOF-assisted Atmospheric Water Harvesting (AWH) offers a cost-effective means of obtaining clean water, even in desert environments. Currently, scientists and engineers have optimized MOF-assisted AWH to produce water at a rate of 3.5 L per kilogram of MOF per day, surpassing the average individual’s daily water needs. The energy consumption for this process stands at 1.67 to 5.25 kilowatt-hours per kilogram of H_2_O at a temperature of 30 °C and a relative humidity of 30% [40]. Additionally, the quality of the water produced meets the standards for direct consumption [40]. Considering an electricity price of $0.07 per kilowatt-hour, the cost of water produced by this system is $0.12 to $0.37 per kilogram of H_2_O. Even when factoring in a carbon tax, such as the current UK rate of $0.1 per kilogram of CO_2_ [75], the cost of harvesting atmospheric water ($0.28 to $0.89 per kilogram of H_2_O) remains below the global average for bottled water ($0.7 per kilogram of H_2_O) (Figure 3).

Given that the MOFs discussed can theoretically raise humidity from 30% to over 80% and considering that the latest minimum energy consumption for 80% industrial AWH is 0.35 kilowatt-hours per kilogram of H_2_O [76], the cost of thermoelectric water production can be reduced to $0.060 per kilogram of H_2_O following engineering optimizations. When utilizing renewable energy sources such as wind power ($0.04), hydroelectricity ($0.02), or intermittent power ($0.01) [77], the cost can be further reduced to $0.0035 to $0.014 per kilogram of H_2_O. Remarkably, this price is comparable to tap water in the United States [78,79], indicating that MOF-assisted AWH, which is not constrained by location or environment, offers the most affordable pure water solution. This technology has the potential to revolutionize the current pipeline-based water supply system and holds vast promise for the future.

### 4.2. Water and Electricity Independent Supplier

By integrating MOF-assisted AWH technology with energy storage solutions, we can construct a decentralized water and power supply system. This innovation will significantly aid humans in mitigating risks and restoring the water cycle. Generally, water-scarce regions are rich in solar and wind energy. The integrated water and power facilities can function as follows: (1) Solar cells and wind turbines harness solar and wind energy, converting it into electricity across expansive arid landscapes, which is then stored in lithium-ion batteries. (2) The assisted AWH system accumulates water from the atmosphere. (3) The stored electrical energy is utilized to facilitate the evaporation of water-saturated MOFs, followed by the collection of condensed water (as depicted in Figure 4).

In the event of an electricity shortfall, the potential energy of water can also be harnessed to generate power. In essence, the entire system operates as a self-sufficient, off-grid water and electricity provider. These systems will not only enhance the quality of life for inhabitants of arid regions [80] but also establish an emergency infrastructure to address the droughts brought on by climate change, thereby bolstering human resilience to risk.

### 4.3. Outlook

MOF-assisted Atmospheric Water Harvesting (AWH) technology facilitates the extraction of water from low-humidity air with minimized energy usage, rendering affordable, purified water globally accessible.

The latest breakthroughs in water-extracting MOFs focus on refining individual MOF materials, chiefly by modifying organic ligands—through functionalization, adjustments in chain length, and incorporating heterocyclic rings with hydrophilic atoms—to precisely control hydrophilicity, water adsorption, and adsorption heat. Looking to the future, the evolution of water-extracting MOFs will pivot toward material compositing, with the use of auxiliary additives to adjust pore size and environment. These composites will be fabricated using a variety of methods, including electrical [81], thermal [82], or doping techniques [83,84,85], to engineer the most effective water extraction materials.

Moreover, as an engineered device, the application of MOFs can be diversified and functionally segmented. For example, a design that integrates a ’low-humidity water adsorption zone’ with a ’high-humidity water storage zone’ can be developed for optimized water extraction. Further engineering optimizations are on the horizon, such as the implementation of fluidized bed systems where MOF particles are suspended in a gas stream to increase cycle rates or the creation of functional polymer composites [86].

In conclusion, with the synergy of academic research, industrial advancements, and governmental funding, MOF-assisted AWH is set to make substantial progress, effectively tackling water scarcity challenges. This technology promises to stabilize urban water supplies, sustain agricultural productivity, support healthy industrial expansion, and maintain ecological equilibrium.

## Figures and Tables

**Figure 1 materials-18-00379-f001:**
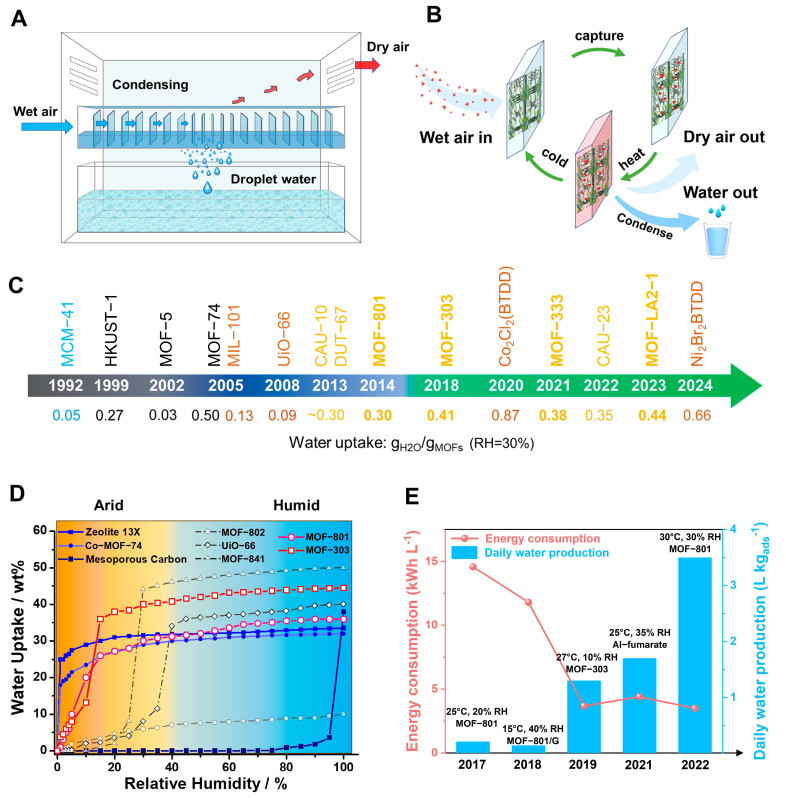
Overview of MOF-assisted AWH with schematic, history, and water uptake properties. (**A**,**B**) Schematic diagram of traditional AWH and the assisted AWH. (**B**) Assisted AWH flow chart. Step 1: porous materials spontaneously adsorb water vapor; Step 2: heat the water-saturated porous materials to liberate water vapor; Step 3: condense vapor to gather water. (**C**) History of water uptake MOFs. Black indicates poor water stability. Yellow indicates that the material can adsorb water below 20% humidity. Brown indicates water adsorption at 20–60% humidity. The water uptake (RH = 30%) of MCM-41 [23], HKUST-1 [24], MOF-5 [25], MOF-74 [26], MIL-101 [26], UiO-66 [26], CAU-10 [27], DUT-67 [28], MOF-801 [28], MOF-303 [29], Co_2_Cl_2_(BTDD) [26], MOF-333 [29], CAU-23 [30], MOF-LA2-1 [20], and Ni_2_Br_2_BTDD [17]. (**D**) Water sorption isothermal at 25 °C of different porous materials. (**E**) In just 5 years, the adsorbent-assisted AWH device has evolved from a gram-level technology validation device and the first on-site water production device to a multi-cycle device that is improving energy consumption [16,31,32,33,34,35,36]. MOF-801 [31], MOF-801/G [37], MOF-303 [38], Al-fumarate [39], and MOF-801 [40]. MOF-801/G is mixed with porous graphite in MOF-801 to enhance its solar energy adsorption performance and thermal physical properties.

**Figure 2 materials-18-00379-f002:**
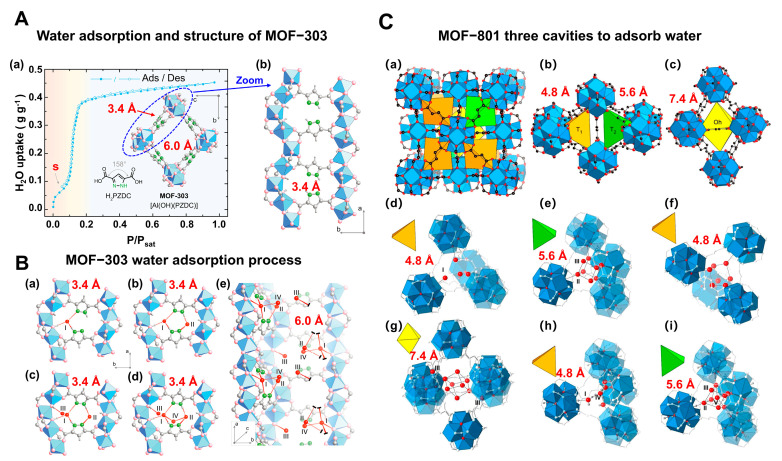
Multi-porous MOF-303 and MOF-801 water adsorption. (**A**) (**a**) Water adsorption isotherm of MOF-303, (**b**) crystal structure characteristics of MOF-303, where Al is represented by blue polyhedrons, O is represented in pink, C and H are represented in gray, and N is represented in green. Here and in other figures, coordinate systems are given for guidance. The three parts of the water adsorption isotherm of MOF-303 are represented in red (forming water clusters), yellow (forming cluster chains), and blue (forming water networks) in the background [29]. Copyright 2021 American Association for the Advancement of Science. (**B**) Schematic diagram of water being gradually adsorbed by MOF-303, where the first water molecule (I) is filled with 3.4 Å diameter pore size until being fully filled by four water molecules (I–IV); the subsequent water molecules overflowed into the 6.0 Å pore, which is perpendicular to 3.4 Å pore. The crystallographic snapshots of MOF-303 adsorbing water molecules were captured at 0.04 g g^−1^ (**a**), 0.06 g g^−1^ (**b**), 0.11 g g^−1^ (**c**), 0.15 g g^−1^ (**d**), and 0.15 g g^−1^ (**e**) [29]. Copyright 2021 American Association for the Advancement of Science. (**C**) (**a**) Schematic diagram of three cavities in MOF-801, (**b**) two tetrahedral, T_1_ and T_2_, which have 4.8Å pores (represented by orange tetrahedrons) and 5.6 Å pores (represented by green tetrahedrons), and (**c**) one octahedral, Oh, which has 7.4 Å pores (represented by yellow tetrahedra). (**d**,**e**) Water firstly stays in tetrahedral cavities with hydrogen-bond formation with the Secondary Building Units (sites I and II), the subsequent water molecules (sites III) form hydrogen bonds with the water molecules at sites II. (**f**) Even the incomplete cubic clusters have the same water adsorption as the fine one. (**g**) Water is also linked by hydrogen bonding in octahedral cavities. (**h**,**i**) The microcrystalline MOF-801-P has a similar sorption sites (I, II, and III), and additional sites IV and V [28]. Copyright 2014 American Chemical Society.

**Figure 3 materials-18-00379-f003:**
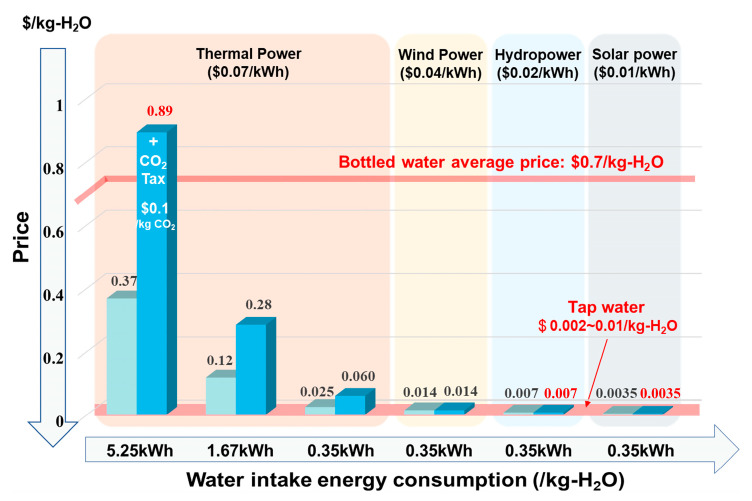
Atmospheric water intake prices when using different forms of energy under various energy consumption levels. The horizontal axis indicates different water intake energy consumption, such as 5.25, 1.67, and 0.35kWh/kg-H_2_O. The four background colors in the figure indicate electricity prices from different sources ranging from $0.07/kWh to $0.01/kWh. The pale aqua column indicates the corresponding water intake price, while verdigris indicates the water intake price after carbon tax. Statement: if the grid power source comes from a thermoelectric source, carbon tax should be considered. Carbon emission of thermal power is approximately 0.86kg-CO_2_/kWh, which assumes that 335 g of standard coal is burned to generate 1 kWh of electricity and that the carbon of the coal is 70%.

**Figure 4 materials-18-00379-f004:**
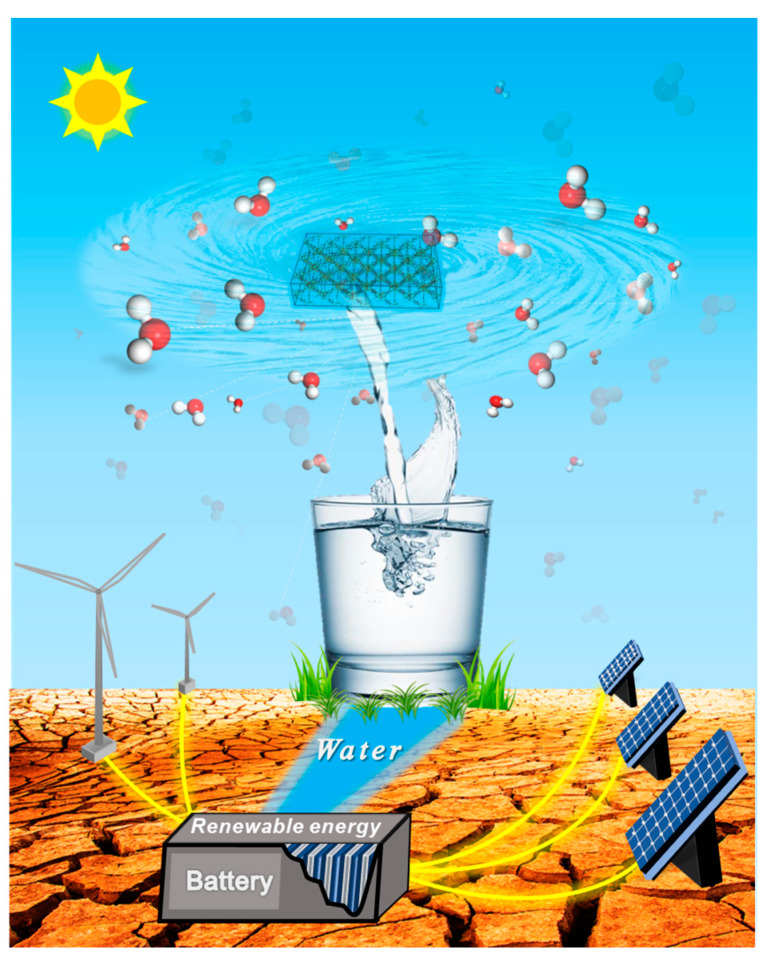
Distributed water and power supply center. Solar and wind provide energy, batteries are responsible for energy storage, and MOF-assisted AWH is used for water extraction, thus constructing a water–electricity decentralized service center.

**Table 1 materials-18-00379-t001:** Details of different types of porous materials.

Classification	H_2_O Bonding Force	Mesoporous Material	Metal Cluster	Organic Ligand	Structure Diagram	Pore Size (Å)	Vp ^c^ (cm^3^ g^−1^)	BET ^b^ (m^2^ g^−1^)	Ref.
Classic inorganic mesoporous materials	weak (Q_st_ = 10 kJ mol^−1^) ^a^	Mesoporous carbon	C-C			10~20	W	1210	[42]
		Mesoporous silicon	Si-O				W		[43]
	Strong (Q_st_ = 70 kJ mol^−1^)	Zeolite-13X	Na_86_[(AlO_2_)_86_(SiO_2_)_106_]•H_2_O		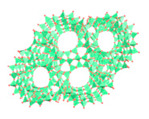	10	0.393	650	[44]
Metal-organic framework compound	Strong (Q_st_ = 70 kJ mol^−1^)	MOF-74-Mg	Mg	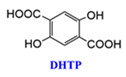	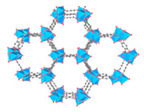	11.1	0.53	1250	[73]
		MOF-74-Co	Co			11.1	0.49	1130	[73]
		MOF-74-Ni	Ni			11.1	0.46	1040	[73]
	Moderate (Q_st_ = 50 kJ mol^−1^)	UiO-66	Zr_6_O_4_(OH)_4_	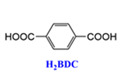	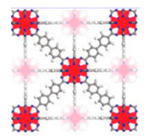	7.4, 8.4	0.49	1290	[59]
		MOF-841	Zr_6_O_4_(OH)_4_	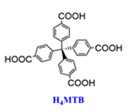	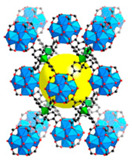	9.2	0.53	1390	[28]
		MOF-802	Zr_6_O_4_(OH)_4_	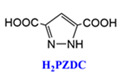	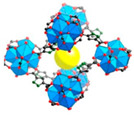	5.6	<0.01	<20	[28]
	Optimum (Q_st_ = 60 kJ mol^−1^)	MOF-801	Zr_6_O_4_(OH)_4_	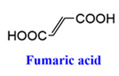	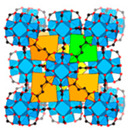	4.8, 5.6, 7.4	0.27	990	[28]
		MOF-303	Al(OH)	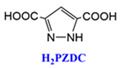	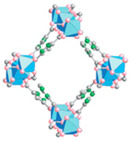	3.4, 6.0	0.58	989	[29]

^a^ Q_st_: isosteric heat of adsorption; ^b^ BET: Brunauer, Emmett, and Teller method to test the material surface area; ^c^ V_p_: pore volume. UiO-66 [59]. Copyright 2008 American Chemical Society. MOF-841, MOF-802, and MOF-801 [28]. Copyright 2014 American Chemical Society. MOF-303 [29]. Copyright 2021 American Association for the Advancement of Science.

## Data Availability

No new data were created or analyzed in this study.

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
