# Peer review of "Metal–Organic Framework-Assisted Atmospheric Water Harvesting Enables Cheap Clean Water Available in an Arid Climate: A Perspective"

_materials, 2025, doi:10.3390/ma18020379_

Round 1

Reviewer 1 Report

Comments and Suggestions for Authors

Materials-3379262

The prospects article reports the rational utilization of Metal-Organic Frameworks (MOFs) in enhanced the Atmospheric Water Harvesting technologies. The review is supported with a lot of schemes, figures, and tables. The article could be accepted after addressing the following comments

1.       The authors should rewrite the abstract to emphasize the novelty and main findings of this prospects.

2.       The authors should add some electron microscope data like TEM, SEM and along with other data

3.       The authors should provide quantitative and qualitative analysis on the summarized data and provide their opinions

4.       In the conclusion sections, the authors should highlight the most promising MOF materials and their preparation methods. Also, they should highlight the current challenges and propose some solutions based on their expertise

5.       A brief comparison between the performance of MOFs and other materials in enhanced the Atmospheric Water Harvesting technologies will be helpful

6.        The authors should revise the references format according to the journal requirements. Some information is missing

7.       The literature review is not enough, and the authors should cite more recent references related to MOF like Langmuir 2022, 38 (36), 11109-11120, Nanoscale Advances 2022, 4 (23), 5044-5055.

Author Response

Comment: The prospects article reports the rational utilization of Metal-Organic Frameworks (MOFs) in enhanced the Atmospheric Water Harvesting technologies. The review is supported with a lot of schemes, figures, and tables. The article could be accepted after addressing the following comments

Key comments:

  1. The authors should rewrite the abstract to emphasize the novelty and main findings of this prospects.

Response to Reviewer#1 comment: We appreciate your comments that the abstract should emphasize the novelty and main findings of this prospects that we shorten the AWH’s technology background and enlighten the standard which MOFs assisted-AWH should meet. Moreover, as this manuscript is Perspective, thus the novelty and main findings are ‘this article gives a concise review of the advancements in enhanced-AWH technologies, delves into the attributes of appropriate MOFs, and offers insights into the potential and future directions of MOFs-AWH. In conclusion, we underscore that that the development of designable MOFs holds the key to widespread practical implementation of AWH, promising the availability of affordable clean water anywhere in the world.’

We rewrite the abstract as below,

Extracting water from atmospheric directly seems perfect way to solve 2 billion people’s water sacrality, however, the traditional Atmospheric Water Harvesting (AWH) lacking ability to adsorb water molecule in arid climate. Porous materials are capable to assist water adsorption, while right now, only some customizable Metal-Organic Frameworks (MOFs) are meet the standard which adsorb water molecules at low humidity and release water at low temperature at some times, which can realize assisted-AWH’s practical and energy-efficient use (Energy consumption<5kWh/L-water). In this perspective, we offer a concise review of the advancements in enhanced-AWH technologies, delves into the attributes of appropriate MOFs, and offers insights into the potential and future directions of MOFs-AWH. In conclusion, we underscore that that the development of designable MOFs holds the key to widespread practical implementation of AWH, promising the availability of affordable clean water anywhere in the world.

  1. The authors should add some electron microscope data like TEM, SEM and along with other data

Response to Reviewer#1 comment: We appreciate your comments which more electron microscope data should be added. However, considering the adsorption and desorption of water molecules are the most crucial steps and water diameter is 3.4Å, in order to detect this which XRD detection is generally used instead of electron microscopy, what’s more SEM images are rarely seen even in research articles.

  1. The authors should provide quantitative and qualitative analysis on the summarized data and provide their opinions

Response to Reviewer#1 comment: Thank you for your guiding comments about summarized data should be analysis and provide opinions.

The data in Figure 1 shows the quantitative analysis which as the time passing by, MOF-801, MOF-303 have been developed. And we add some comment to provide some opinions.

Page 4: “To date, MOF-801 and MOF-303 have been the subjects of extensive research for they have excellent water absorption characteristics at lower humidity and have large water absorption capacity (Figure 1C and 1D)”

The data in Figure 2 shows the qualitative analysis which ‘AWH’s MOF should have 3.4Å pores which contented five hydrogen bonds to make adsorb water molecules at low humidity. When adsorbed water molecules aggregate, a larger space is required to carry them, such as 6Å pores.’

For make this statement clearer, we have made some changes in articles, 

Page 7: “Water contains hydrogen bonds. When MOF’s pores have hydrogen bonds and their diameter is similar to the water molecules (3.4Å), that individual water molecule can be captured easily”

Moreover, the data in Table 1 shows the qualitative analysis which the relationship between isosteric heat of adsorption (Qst) and water adsorption.

  1. In the conclusion sections, the authors should highlight the most promising MOF materials and their preparation methods. Also, they should highlight the current challenges and propose some solutions based on their expertise

Response to Reviewer#1 comment: We appreciate your comments, we do think we have highlight MOF-303 and MOF-801 are the most promising MOF materials for MOFs-AWH. Considering this manuscript is Perspective that should be more forward looking and speculative than Reviews, so materials preparation methods are not mentioned in the article. Moreover, there are series reference listed below that contain preparation methods, such as Materials Today 2024, 78, 92-111. J Am Chem Soc 2014, 136 (11), 4369-81. ACS Central Science 2020, 6, 1348−1354.

For an advanced device that is about to be applied in the market, the challenges come from multiple aspects, including re-optimization and re-innovation of water absorbing materials, device combination methods involving fluid mechanics and water absorbing material stacking, and the repetitive process of water intake and discharge. Therefore, the detection and logical feedback of device operation also need to be considered. These points are mentioned in Page 12 'OUTLOOK AND PERSPECTIVE' and more detailed can be obtained from our published review Materials Today 2024, 78, 92-111.

What’s more, we shall worth noting that this article is published as Perspective, which aims to enable more people easily understand the value of MOF-AWH in development, its development overview, future application costs, future application scenarios, and areas for iterative improvement, rather than detailed Review or Research Article.

  1. A brief comparison between the performance of MOFs and other materials in enhanced the Atmospheric Water Harvesting technologies will be helpful

Response to Reviewer#1 comment: Thank you for your comment, that the comparison between MOF and other materials are detailed description in 'searching the suitable MOFs' and compared it in Figure 1D.

For make this statement clearer, we have made some changes in articles, 

Page 4: “To date, MOF-801 and MOF-303 have been the subjects of extensive research for they have excellent water absorption characteristics at lower humidity and have large water absorption capacity (Figure 1C and 1D)”

  1. The authors should revise the references format according to the journal requirements. Some information is missing

Response to Reviewer#1 comment: Thanks for your careful review. We have carefully checked and revised the format of the references in accordance with the requirements of the journal, and supplemented some missing information in the references.

  1. The literature review is not enough, and the authors should cite more recent references related to MOF like Langmuir 2022, 38 (36), 11109-11120, Nanoscale Advances 2022, 4 (23), 5044-5055.

Response to Reviewer#1 comment: We appreciate your comments that more effective references should be added.

As you know, there are countless MOF materials, and their applications vary greatly. The core of this article is the use of MOFs with special designs and structures for water capture and release in arid regions. Even though our team has published many MOF articles, they were not cited here as they are not related to water extraction. The articles proposed by the reviewers are mostly laboratory electrocatalytic articles, and the MOF materials and sintered materials mentioned in the article are not suitable for water extraction. After consideration, this article will not be cited here.

When we publish catalytic articles, we shall continue to pay attention to the articles you mentioned and cite them

Reviewer 2 Report

Comments and Suggestions for Authors

This perspective concisely reviews the advancements in enhanced atmospheric water harvesting technologies using appropriate metal-organic frameworks. However, I recommend a minor revision before acceptance.

1— I recommend that the authors modify Section 3, "Characteristics of Suitable MOFs," to focus first on MOF properties and then on "Facing the Challenges."

2—The authors should consider the topic's limitations, future scope, and outlook and make necessary modifications to enhance clarity.

3— Minor editing of the English language is required.

Comments on the Quality of English Language

Minor editing.

Author Response

Comment: This perspective concisely reviews the advancements in enhanced atmospheric water harvesting technologies using appropriate metal-organic frameworks. However, I recommend a minor revision before acceptance.

Key comments:

  1. I recommend that the authors modify Section 3, "Characteristics of Suitable MOFs," to focus first on MOF properties and then on "Facing the Challenges."

Response to Reviewer#2 comment: We are grateful to the reviewer comments about description of MOF properties firstly, that we have added it on page 6.

Page 6 :“ MOF properties

MOFs are a class of porous materials composed of metal nodes and organic linkers. Their modular nature allows for great synthetic tunability, affording both fine chemical and structural control. With creative synthetic design, properties such as porosity, stability, porous size, porous environment especially chemical and physical bond force can be tailored for specific applications. The pore size and pore environment in MOFs play an important role in the adsorption and desorption process of water. For low humidity conditions, capturing sparse water molecules requires pores that are close in size to water molecules and have hydrogen bonding adsorption capabilities. When water molecules aggregate and their volume increases, large pores are needed to accommodate them.

  1. The authors should consider the topic's limitations, future scope, and outlook and make necessary modifications to enhance clarity.

Response to Reviewer#2 comment: We appreciate your comments that we add some comment to enhance the clarity.

Page 12: “persorward to MOFs assisted AWH from three aspects: water intake cost, usage scenarios, and areas for improvement in the future.”

  1. Minor editing of the English language is required.

Response to Reviewer#2 comment: Thanks for your careful review. We have carefully checked and refine this article.

Reviewer 3 Report

Comments and Suggestions for Authors

This review focused on the application of Metal-Organic Frameworks (MOFs) for Atmospheric Water Harvesting. In general, the authors include an easy-reading document, and the organization is very appropriate to this Journal. In my opinion, this document could be a guide for other researchers that want to start the study on this application; thus, I recommend to revise the document before publication. Please, considering the following points:

1.- Include more perspective and conclusions in the abstract. It is very general. The authors could include the intervals of the water capacity capture of the most studied MOFs, to have a general idea of the real capacity of these materials.

2.- Section 2.1. Include a scheme to explain the different steps in the mechanism.

3.- Figure 1c. It could be a good idea to include the capture efficiency of each MOF shown in the timeline. To see the evolution of the efficiencies over time.

4.- Page 4, line 127-128. Specify what is "small pore".  Include the interval of radious/diameter of the porosity.

5.- Section 3.3. It is not enough clear the functionalization of MOFs. Include specific information of what methodologies/strategies have been implemented; also, what type of compounds are generally used to this application.

6.- Is there any relation of the pore volume with the water capacity of the materials?

7.- Section 4. This part should be included other analysis of the risk of using these compounds, which toxicity is still under debate (see https://doi.org/10.1039/D1CS00918D)

8.- Include a separate section of conclusions or outlook and perspectives of application. (Expand this latter section).

Other comments

9.- Revise the format; some references appear in red.

10.- Revise grammar in some sentences, e.g., "this perspective zeroes...".

Author Response

Comment: This review focused on the application of Metal-Organic Frameworks (MOFs) for Atmospheric Water Harvesting. In general, the authors include an easy-reading document, and the organization is very appropriate to this Journal. In my opinion, this document could be a guide for other researchers that want to start the study on this application; thus, I recommend to revise the document before publication. Please, considering the following points:

Key comments:

  1. Include more perspective and conclusions in the abstract. It is very general. The authors could include the intervals of the water capacity capture of the most studied MOFs, to have a general idea of the real capacity of these materials.

Response to Reviewer#3 comment: We are grateful to your comments that water capacity should be pointed out in the abstract.

As depicted in Figure 1D, the water uptake values of most studied MOFs are between 30wt% to 40wt%, but it changes with relative humidity. For MOFs assisted AWH, the most important things are energy consumption and daily water production, those are industry index, not only related to water absorbing materials, but also to the environment, materials packing, engineering strategies, etc.

What’s more, in order to emphasize the novelty, we rewrite the abstract as below,

Extracting water from atmospheric directly seems perfect way to solve 2 billion people’s water sacrality, however, the traditional Atmospheric Water Harvesting (AWH) lacking ability to adsorb water molecule in arid climate. Porous materials are capable to assist water adsorption, while right now, only some customizable Metal-Organic Frameworks (MOFs) are meet the standard which adsorb water molecules at low humidity and release water at low temperature at some times, which can realize assisted-AWH’s practical and energy-efficient use (Energy consumption<5kWh/L-water). In this perspective, we offer a concise review of the advancements in enhanced-AWH technologies, delves into the attributes of appropriate MOFs, and offers insights into the potential and future directions of MOFs-AWH. In conclusion, we underscore that that the development of designable MOFs holds the key to widespread practical implementation of AWH, promising the availability of affordable clean water anywhere in the world.

  1. Section 2.1. Include a scheme to explain the different steps in the mechanism.

Response to Reviewer#3 comment: We appreciate your comments that it should have scheme to explain the different steps in the mechanism. However, we think we have made it in the Figure 1A and Figure 1B, maybe it a bit of small to be read, thus we have added some comment on page 5.

Page 5: ‘(B) Step 1: Porous materials spontaneously absorb water vapor; Step 2: Heat the water-saturated porous materials to liberate water vapor; Step 3: condense vapor to gather water.’

  1. Figure 1c. It could be a good idea to include the capture efficiency of each MOF shown in the timeline. To see the evolution of the efficiencies over time.

Response to Reviewer#3 comment: Thank you for your valuable suggestions, which are of great benefit to modifying our manuscript. We have carefully revised the Figure 1c by adding the water uptake capacity of each MOF at RH=30%. This indicator reflects the capture efficiency of MOF for water vapor and is a key indicator determining the adsorption performance of AWH.

  1. Page 4, line 127-128. Specify what is "small pore".  Include the interval of radious/diameter of the porosity.

Response to Reviewer#3 comment:

We are grateful to the reviewer comments to specify the ‘small pore’that we have added some comment on page 7.

Page 7: ‘The small pores (diameter nearly to 3.4Å) exhibit strong interactions with individual water molecules, enabling the capture of water at low relative humidity.’

  1. Section 3.3. It is not enough clear the functionalization of MOFs. Include specific information of what methodologies/strategies have been implemented; also, what type of compounds are generally used to this application.

Response to Reviewer#3 comment: We are grateful to the reviewer comments about the functionalization of MOFs criteria. Two things are count for the functionalization of MOFs, one is pore size which appropriate between a single water molecule (3.4Å) and a water cluster (6-8.0Å), the other is pore environment which provided by ligands with functional groups such as N, O, F, etc. that have hydrogen bonding structures with water molecules.

In order to clarify it well, we have changed some comment on Page 7.

Page 7: ‘Water contains a large number of  hydrogen bonds73, 74. When MOF’s pores have hydrogen bonds and their diameter is similar to the water molecules (3.4Å), that individual water molecule can be captured easily. The volume density of hydrogen bonds—the number and proximity of hydrogen bonds within the pores—significantly influences the adsorption and storage of water. The count of hydrogen bonds can be manipulated by altering the prevalence of nitrogen, oxygen, and fluorine elements within the organic ligands that serve as the connectors in MOFs. Additionally, the distance between hydrogen bonds can be adjusted by modifying the length of the organic ligands.’

  1. Is there any relation of the pore volume with the water capacity of the materials?

Response to Reviewer#3 comment: Frankly, there is no relationship between pore volume and water capacity. A large pore volume does not necessarily gain a higher water absorption capacity, for it is also depended on the interaction force between pore and water. But to the small pore volume, and in other similar situations, the water capacity is absolutely small.

  1. Section 4. This part should be included other analysis of the risk of using these compounds, which toxicity is still under debate (see https://doi.org/10.1039/D1CS00918D)

Response to Reviewer#3 comment: Thanks to your valuable suggestion that safety and health issues are the primary concerns in technological development.

Nanoparticles entering cells, tissues, etc. should be clinically studied to verify their toxicity. It is worth noting that the core of MOF materials for water extraction is that they are not easily hydrolyzed. According to reference 58 (Nature Communications 2022, 13, 4873), after MOF-303 was prepared into a water extraction device, no metal ions were detected by chromatographic analysis during one year of operation, and the water cleanliness remained at the level of direct drinking water in Jordan. There is no toxicity issue at this point.

When it comes to whether there are toxicity issues when preparing MOF materials for water extraction into devices, we believe the industrial assembly robots may completely avoid such problems.

Page 12,we marked into red to show this statement,  “Additionally, the quality of the water produced meets the standards for direct consumption[58]”

  1. Include a separate section of conclusions or outlook and perspectives of application. (Expand this latter section).

Response to comment: We are sorry for we haven’t got your meaning.

This Perspective aims to enable more people easily understand the value of MOF-AWH in development, its development overview, future application costs, future application scenarios, and areas for iterative improvement, rather than detailed Review or Research Article.

To make it clearly, we add some comment to enhance the clarity in Page 12.

Page 12: “We will look forward to MOFs assisted AWH from three aspects: water intake cost, usage scenarios, and areas for improvement in the future.”

  1. Revise the format; some references appear in red.

Response to Reviewer#3 comment: Thank you for pointing out the formatting issue in the manuscript. We have strictly and carefully revised the format of the main text and references, and changed the references displayed in red to black.

  1. Revise grammar in some sentences, e.g., "this perspective zeroes...".

Response to Reviewer#3 comment: Thank you for pointing out this mistake that we correct it on Page 3.

Page 3: “this perspective focus on… ”

Reviewer 4 Report

Comments and Suggestions for Authors

Reviewer report on materials-3379262

Yang Hu et al.Metal-Organic Frameworks Assisted-Atmospheric Water Harvesting Enable Cheap Clean Water Available in Arid: A Perspective

Atmospheric Water Harvesting (AWH) technology, which extracts water from the air, offers a solution to the global water scarcity challenge. However, conventional AWH is heavily reliant on environmental humidity and is energy-inefficient, making its application in arid regions— where clean water is critically needed—largely impractical. The enhanced-AWH overcomes this limitation by incorporating self-adsorbing porous materials that enable water extraction at low humidity levels. Currently, among the various sorbent materials available, only porous, designable Metal-Organic Frameworks (MOFs) are suitable for practical, energy-efficient application. In this perspective, the manuscript provides a concise review of the advancements in enhanced-AWH technologies, delves into the attributes of appropriate MOFs, and offers insights into the potential and future directions of MOFs-AWH. It concludes by emphasizing that the development of designable MOFs holds the key to widespread practical implementation of AWH, promising the availability of affordable clean water anywhere in the world.

The quality of the manuscript is good. Therefore, the manuscript can be accepted in present form.

Author Response

Comment: Atmospheric Water Harvesting (AWH) technology, which extracts water from the air, offers a solution to the global water scarcity challenge. However, conventional AWH is heavily reliant on environmental humidity and is energy-inefficient, making its application in arid regions— where clean water is critically needed—largely impractical. The enhanced-AWH overcomes this limitation by incorporating self-adsorbing porous materials that enable water extraction at low humidity levels. Currently, among the various sorbent materials available, only porous, designable Metal-Organic Frameworks (MOFs) are suitable for practical, energy-efficient application. In this perspective, the manuscript provides a concise review of the advancements in enhanced-AWH technologies, delves into the attributes of appropriate MOFs, and offers insights into the potential and future directions of MOFs-AWH. It concludes by emphasizing that the development of designable MOFs holds the key to widespread practical implementation of AWH, promising the availability of affordable clean water anywhere in the world.

The quality of the manuscript is good. Therefore, the manuscript can be accepted in present form.

Response to Reviewer#4 comment: We are grateful to the reviewer comments.